# p53 Dysregulation in Breast Cancer: Insights on Mutations in the *TP53* Network and p53 Isoform Expression

**DOI:** 10.3390/ijms241210078

**Published:** 2023-06-13

**Authors:** Luiza Steffens Reinhardt, Kira Groen, Alexandre Xavier, Kelly A. Avery-Kiejda

**Affiliations:** 1School of Biomedical Sciences and Pharmacy, College of Health, Medicine and Wellbeing, The University of Newcastle, Callaghan, NSW 2308, Australia; luiza.steffens@newcastle.edu.au (L.S.R.); kira.groen@newcastle.edu.au (K.G.); alexandre.xavier@newcastle.edu.au (A.X.); 2Hunter Medical Research Institute, New Lambton Heights, NSW 2305, Australia; 3Cancer Detection & Therapy Research Program, Hunter Medical Research Institute, New Lambton Heights, NSW 2305, Australia

**Keywords:** *TP53*, p53 isoform, breast cancer, next-generation sequencing, immunohistochemistry

## Abstract

In breast cancer, p53 expression levels are better predictors of outcome and chemotherapy response than *TP53* mutation. Several molecular mechanisms that modulate p53 levels and functions, including p53 isoform expression, have been described, and may contribute to deregulated p53 activities and worse cancer outcomes. In this study, *TP53* and regulators of the p53 pathway were sequenced by targeted next-generation sequencing in a cohort of 137 invasive ductal carcinomas and associations between the identified sequence variants, and p53 and p53 isoform expression were explored. The results demonstrate significant variability in levels of p53 isoform expression and *TP53* variant types among tumours. We have shown that *TP53* truncating and missense mutations modulate p53 levels. Further, intronic mutations, particularly polymorphisms in intron 4, which can affect the translation from the internal *TP53* promoter, were associated with increased Δ133p53 levels. Differential expression of p53 and p53 isoforms was associated with the enrichment of sequence variants in p53 interactors *BRCA1*, *PALB2,* and *CHEK2*. Taken together, these results underpin the complexity of p53 and p53 isoform regulation. Furthermore, given the growing evidence associating dysregulated levels of p53 isoforms with cancer progression, certain *TP53* sequence variants that show strong links to p53 isoform expression may advance the field of prognostic biomarker study in breast cancer.

## 1. Introduction

The tumour suppressor *TP53* is highly mutated in cancer [1], however, in breast cancer, *TP53* mutation status is highly variable, ranging from approximately 10% in Luminal A patients to 80% in triple-negative breast cancers (TNBCs) [2]. Several studies have demonstrated associations between mutations in *TP53* and clinicopathological features, as well as worse prognosis and resistance to therapies [1,3,4,5,6,7]. Yet, associations of p53 status with positive or neutral outcomes have also been described [4]. These conflicting results are most likely due to differences in the analysis of p53 status and variability within cohorts (i.e., stratification according to hormone status, therapies and cancer subtypes) and considerably impact the clinical significance of p53 status [4].

It has been suggested that p53 expression levels in breast cancer are better predictors of outcome and chemotherapy response than *TP53* mutation [8,9,10]. For instance, the immunohistochemical evaluation of p53 expression patterns may be a prognostic factor in TNBC where high p53 levels predict worse prognosis [11,12,13]; nevertheless, no standardised prognostic test for p53 status, for either mutations or expression, has been proposed. *TP53* mutations may affect p53 expression [2,14], yet other molecular mechanisms can also modulate p53 levels, hence its canonical roles. These molecular mechanisms include loss of function of ATM-CHEK2-p53 signalling [15], *HDM2/4* amplification or activation [16,17,18], *CDKN2A* alterations [19,20] and aberrant expression of long non-coding RNAs [21,22]. Moreover, we and others have demonstrated that p53 isoform expression contributes to deregulated p53 activities and may reveal novel avenues for cancer prognostication [23,24,25,26,27,28,29,30,31,32,33,34,35,36,37,38,39,40,41,42,43,44,45,46,47,48,49,50,51,52,53,54].

p53 is expressed as the full-length protein (p53α) as well as N-terminally (p53β and p53γ), C-terminally (Δ40p53, Δ133p53 and Δ160p53), or N- and C-terminally (Δ40p53β, Δ40p53γ, Δ133p53β, Δ133p53γ, Δ160p53β and Δ160p53γ) truncated isoforms [34,49,55]. These isoforms are produced through different mechanisms including alternative splicing, alternative promoter usage and alternative initiation of translation, as well as post-translational degradation of p53 [34,55,56,57,58,59,60]. Likewise, p53 isoform expression may be fine-tuned by epigenetic regulation of *TP53* promoters [34] and polymorphisms that affect *TP53* promoters such as R72P or PIN3 [50,54,61,62].

Although there are a number of openly available datasets on whole exome sequencing of breast cancers [3], intronic mutations that create cryptic splice sites, mutations within the internal or proximal promoter, and other rarer variants cannot be explored in such datasets. Since such mutations may affect p53 transcription and hence, the expression of p53 isoforms [63], in this study, *TP53* and regulators of the p53 pathway were sequenced by targeted next-generation sequencing (NGS) and associations between the identified sequence variants and p53 and p53 isoform expression were investigated. The results indicate that p53 levels predict breast cancer outcomes and diagnosis and are associated with *TP53* truncating and missense mutations. Differential p53 isoform levels may be modulated by sequence variants in *TP53*, particularly polymorphisms in intron 4 that may regulate Δ133p53 levels and were associated with alterations in genes of key p53 interactors: *BRCA1*, *PALB2*, and *CHEK2.*

## 2. Results

### 2.1. The Mutation Landscape of TP53 in Breast Cancers with Variable Levels of p53

In our previous study, we have shown a trend towards worse disease-free survival in breast tumours with weak and strong DO-1 staining (p53 antibody capable of detecting all isoforms retaining the transactivation domain, TAp53, and used as a surrogate for full-length p53 detection) compared to moderate levels [64]. This was in agreement with another investigation in a larger breast cancer cohort [65]. Therefore, in this study, we evaluated if clinical features correlate with p53 levels previously analysed by IHC in 108 breast cancers [64], comprising 31 Grade 1, 24 Grade 2, and 53 Grade-invasive ductal carcinomas (IDCs) [64]. Significant associations between levels of p53, tumour size and TNBC subtype were observed (Figure 1A,B), with tumours expressing moderate p53 showing decreased tumour size and no triple-negative phenotype. This strongly suggests that the appropriate levels of p53 are important for breast cancer prognosis.

In breast cancer, increased expression of p53 has been associated with worse prognoses [10] and p53 status, since mutant p53 may be more stable than wild-type p53 [66]. Hence, to investigate *TP53* mutations in breast cancers with varying levels of p53, we next sequenced *TP53* in a larger cohort of IDCs (*n* = 137), comprising 34 Grade 1, 34 Grade 2, and 69 Grade 3 cancers. The median age at diagnosis was 55 years old (range 28–90 years old) and the majority of samples were oestrogen receptor (ER)-positive (75.9%), whereas around 13.1% of samples were TNBCs. The libraries were pooled and sequenced using a custom, targeted NGS approach including introns, exons, promoters and untranslated regions (UTRs).

When including all sequence variants, the large majority of variants were found in introns, with single nucleotide variants (SNVs) and deletions being the most predominant classes (Appendix A). By filtering the silent sequence variants (*n* = 2988) using MAF tools [67], mutations were detected in 91% of tumours (median of one variant per sample across the cohort), with SNVs being by far the most predominant variant type (179 SNVs, 8 deletions, and 5 insertions). The variant allele frequency (VAF) varied considerably among variants, but the majority of *TP53* sequence variants presented a VAF of around 0.2 (Figure 1C). Interestingly, almost all tumours presented an intronic substitution at a splice site in intron 5 (c.559+2T>G), which has unknown mutation outcomes (COSMIC ID COSV52900265) but has been classified as likely pathogenic in VarSome [68]. This sequence variant was found in the majority of samples (115/137) and does not affect p53 levels or patient disease-free survival compared to non-carriers in our cohort (Appendix A). When filtering out this sequence variant, *TP53* variants were found in 49% of tumours (Figure 1D). The majority of the remaining sequence variants were either missense or nonsense mutations, followed by frameshift alterations (Figure 1E). The most predominant SNV classes were found to be C>T and T>G (Figure 1F). As expected [69], the majority of mutations were found within the DNA-binding domain of p53 (Figure 1G).

No changes in disease-free survival were observed between samples harbouring *TP53* variants compared to non-carriers (Figure 1H). However, *TP53* sequence variants were enriched in ER- (*p* = 0.0001) and PR-negative tumours (*p* = 0.0048) (Figure 1I), which is expected given the associations between p53 mutation status and TNBCs [2].

To determine whether certain *TP53* mutations affect p53 levels, we matched *TP53* mutations with IHC levels of p53 (*n* = 101). By selecting only tumours that harbour high-impact sequence variants (known pathogenic variants or most likely pathogenic variants assumed to have a disruptive impact on the protein) (Table 1) and matching them with the IHC-staining of p53, it was observed that the majority of sequence variants were found in weak or strong p53-staining IDCs (Figure 2). When comparing to all other tumours, strong and weak-expressing tumours were enriched in cases harbouring pathogenic variants (Appendix A, *p* < 0.0001). Stop-gained and frameshift mutations were largely found in weak-expressing tumours, whereas missense mutations were highly predominant among strong-expressing tumours (Figure 2). Moreover, the majority of high-impact *TP53* variants were exclusive to individual tumours, with the exception of c.574C>T and c.1024C>T, which were found in two tumours.

### 2.2. TP53 Mutations and p53 Isoform Expression

We next evaluated if *TP53* sequence variants could be associated with altered levels of p53 isoforms. The previously analysed H-scores of TAp53, p53β, Δ133p53 and Δ160p53 [64] were used and their relationship with *TP53* sequence variants was evaluated. When looking at mutations detected in exons, high variability in p53 isoform expression and variant types were observed (Figure 3A). Tumours harbouring mutations in exon 4, particularly a missense mutation (c.239C>T) in tumour 109 (which also harbours a frameshift c.797del alteration in exon 8), exhibited increased levels of p53 isoforms, especially Δ133p53, relative to TAp53. A similar expression pattern was observed in tumour 107, which harbours a frameshift alteration (c.406del) in exon 5, and tumours 104, 108 and 120, which exhibit alterations in exon 8 (in-frame variant c.838_846dup, frameshift variant c.846dup and frameshift variant c.902dup, respectively). Moreover, tumour 123 presented an in-frame deletion (c.529_546del) in exon 5 and strong levels of TAp53, but low levels of p53 isoforms (Figure 3A). Other sequence variants were also detected in the 3′UTR and 5′UTR of *TP53*.

Out of the 137 IDCs, all but two harboured *TP53* hotspot mutations (tumour 125: R175H and tumour 118: R273L; Table 1). p53 expression pattern was found to be strong in these specimens (Figure 2) and when looking at the p53 isoform expression, increased TAp53 and Δ160p53 H-scores were observed in both samples compared to the median H-scores for each isoform (Appendix A). p53β was expressed at lower levels in both samples, whereas Δ133p53 was expressed at increased levels in tumour 118 and lower levels in tumour 125 (which also harbours a novel intronic variant c.375+14A>C in intron 4) (Appendix A).

A large amount of missense variants were detected in exon 7, which is expected since mutations in the DNA-binding domain of *TP53* are predominantly missense [70]. Both p53 and its isoforms showed highly variable H-scores in tumours harbouring variants in exon 7 (Figure 3B). When looking at specific mutations in exon 7, c.776A>C (p.Asp259Ala) was highly prevalent mainly in tumours with weak TAp53 levels (Figure 3C). By examining the tumours that exhibited this missense variant and moderate or high TAp53 levels (black circle in Figure 3C), it was observed that these tumours also harbour other variants such as missense, frameshift, splice acceptor (Figure 3D) and intronic (Appendix A) variants, whereas weak TAp53 tumours only displayed additional intronic variants (Appendix A). When looking at only weak TAp53 cases, increased levels of p53β and Δ133p53 were observed compared to TAp53 (Figure 3E), suggesting that the levels of p53 isoform may not correlate with p53 expression, even in samples that harbour the same mutations.

Cases harbouring stop-gained mutations generally showed low expression of all p53 isoforms, with the exception of tumour 112, which harbours a c.637C>T stop-gained mutation and increased levels of p53β and Δ133p53 (Figure 3F). It should be noted that this tumour only exhibited additional 3′ flank and intronic variants (Appendix A).

As intronic variants were detected in all tumours (Appendix A), several variants were highly prevalent, and the majority of variants were either found in almost all cases or in just a few samples. We have focused on intronic sites that could potentially influence p53 isoform generation, particularly sequences affecting splicing. Only five tumours exhibited sequence variants in splice sites and the p53 isoform levels varied considerably among those samples (Figure 3G). Two tumours harbouring splice site variants located in intron 4 (splice acceptor: c.376-1G>A) and 9 (splice region: c.994-8T>A) presented high TAp53 levels and low p53 isoform levels, whereas two other tumours showed increased levels of p53β or Δ133p53 and variants in intron 6 (splice acceptor: c.673-1G>A) or 9 (splice acceptor: c.994-1G>A), respectively. One case exhibited low levels of all p53 isoforms and a variant in intron 5 (splice donor: c.559+1G>T).

Among other variants found in introns, tumours harbouring a c.376-125T>C variant in intron 4 showed low TAp53 and Δ160p53, variable p53β and high Δ133p53 levels, except for tumour 21 (which also harbours the missense variant c.725_726delinsTT) and tumour 113 (which also harbours the missense and high impact variant c.659A>G) (Figure 3H). A similar expression pattern was also observed in tumour 109 (Figure 3H), which harbours a c.376-117G>A variant in intron 4 and other variants such as c.239C>T and c.797del (Figure 3A). We also identified other sequence variants in intron 4 (c.375+14A>C, c.376-91G>A and c.376-86T>C), but no associations were detected. These results suggest that sequence variants found in intron 4 may be associated with altered levels of p53 isoforms.

To examine if a specific mutation type or sequence variant was enriched when distinct p53 isoforms were expressed individually or in combination (given that p53 isoforms would be normally expressed in combination), the p53 isoform (TAp53, p53β, Δ133p53 and Δ160p53) H-scores were segregated into low (L) or high (H) levels using the median expressions as cut-off values. No significant results were found for frameshift, nonsense and intronic mutation types, however, missense mutations were enriched in high TAp53 cases (*p* < 0.0001) and cases with high TAp53 and Δ133p53 and low p53β and Δ160p53 (HLHL) (*p* = 0.0362) or with high TAp53 and Δ160p53 and low p53β and Δ133p53 (HLLH) (*p* = 0.0428) (Figure 4A).

### 2.3. Alterations in p53 Interactors and p53 Isoform Expression

Knowing that alterations in genes associated with the p53 pathway can modulate not only p53 functions but also its levels [15,16,17,19,20], enrichment analyses of sequence variants in p53 interactors (see Methods for full list) were performed in tumours with varying levels of p53 isoforms. Sequence variants in *BRCA1* were enriched in high TAp53 (*p* = 0.0034), whereas *PALB2* variants were enriched in low TAp53-expressing cases (*p* = 0.0387) (Figure 4B). Similarly, enrichment of *BRCA1* sequence variants was observed in high TAp53 and Δ160p53, and low p53β and Δ133p53 tumours (HLLH) (*p* = 0. 0068) (Figure 4B). For individual isoforms, the only statistically significant result was found for high p53β cases, which showed enrichment of *CHEK2* sequence variants (*p* = 0.0476) (Figure 4B). When looking at specific sequence variants found in *BRCA1*, *CHEK2* and *PALB2*, novel variants were detected for all three genes (Table 2). These variants were predominantly specific to individual tumours, except for missense mutations detected in *PALB2* (c.1651T>A; p.Tyr551Asn) and in *CHEK2* (c.1261A>C; p.Thr421Pro), which were observed in 13 and 130 specimens, respectively, resulting in a high mutation burden, particularly for *CHEK2*. Taken together, these results indicate a relationship between the levels of p53 and its isoforms and genetic alterations of p53-related genes, however, additional studies are required to assess the impact of novel mutations and the outcomes of these associations.

## 3. Discussion

In this study, we demonstrated that *TP53* mutations modulate p53 levels and may affect the generation of p53 isoforms. p53 levels predict breast cancer outcomes and prognosis. Furthermore, the levels of p53 and its isoforms were associated with the enrichment of genetic alterations in p53 interactors *BRCA1*, *PALB2* and *CHEK2*, which are genes with known impact on breast cancer outcomes.

We and others have underpinned the association between dysregulated levels of p53 and breast cancer prognosis [64,65]. Here, we have reinforced this hypothesis and shown an association between p53 levels and enrichment of truncating and missense mutations in *TP53* and homologous recombination repair-related genes: *BRCA1* (when p53 is highly expressed), and *PALB2* (when p53 is weakly expressed or absent) (Figure 4). Since *BRCA1* and *PALB2* pathogenic variants are related to breast cancer predisposition, adverse clinical prognosis, and aggressive clinical features [71,72,73], these findings further support the association between p53-dysregulated levels, worse disease-free survival [64], and clinical features such as tumour size, triple-negative phenotype, and hormone receptor status (Figure 1). It has been observed that the vast majority of breast cancers mutant for *BRCA1* also harbour *TP53* mutations [74]. Moreover, somatic p53 loss-of-function or dysfunction may accelerate BRCA- and PALB2-associated tumorigenesis [75]. Altogether, the investigation of p53 functionality in *BRCA1* and *PALB2* mutant cancers may raise novel prospects for combination therapies against breast cancers with homologous recombination deficiency by targeting p53-dysregulated cells and BRCA1-deficiency using target therapies.

Even though the evaluation of p53 levels may help in taming the currently stormy relationship between breast cancer prognostication and p53 status [4], our results showed that p53 is heterogeneously expressed in IDCs (Figure 2). This is likely due to the array of patient-specific mutations and epigenetic factors and highlights the significant complexity of p53 regulation. This complexity is even more evident when including the expression of p53 isoforms, which can be detrimental or beneficial to p53′s canonical functions in a context-specific fashion (reviewed in [49,76,77,78]). In this study, only two tumours presented *TP53* hotspot mutations (tumour 125: R175H and tumour 118: R273L; Table 1) and increased p53 and Δ160p53 levels were observed in both samples (Appendix A) compared to the median H-scores for these isoforms. These findings are in accordance with a previous study, which demonstrated that hotspot mutations in *TP53* may lead to increased Δ160p53 levels compared to cells harbouring wild-type *TP53* [79] and that the gain-of-functions of mutant p53 may be explained to some extent by the expression of Δ160p53 [79]. Nevertheless, compared to full-length p53, Δ160p53 was expressed at lower levels [79], similar to our findings [79]. Interestingly, in both tumours, p53β was expressed at low levels, contrasting to Δ133p53, which was expressed at low levels in the specimen harbouring the R175H hotspot (tumour 125), but at high levels in the specimen harbouring the R273L mutation (tumour 118). This could be explained by a mutation in intron 4 (internal promoter of *TP53*) found in tumour 125, which could potentially affect the translation of Δ133p53 (discussed below). These observations further underpin the complexity of p53 isoform levels regulation; however, functional assays are needed to test the suggested associations.

Our results clearly demonstrate the significant variability in levels of p53 isoforms and variant types among tumours (Figure 3). For instance, even though stop-gained mutations indicated a more consistent expression pattern (low expression of all p53 isoforms), tumour 112, which harbours a known pathogenic *TP53* mutation: c.637C>T (rs397516436), presented increased levels of p53β and Δ133p53 (Figure 3F). This tumour also harboured SNVs and deletions in introns 1, 3, 6 and 3′ flank regions, which could influence 3′ splicing to favour the generation of p53β variants [50] (in this case p53β and Δ133p53β, which are both detected by the antibodies used in the IHC). Tumours harbouring other variants located in splice sites in intron 4 (splice acceptor: c.376-1G>A) and 9 (splice region: c.994-8T>A) presented high TAp53 levels and low p53 isoform levels (Figure 3G). These variants could possibly modulate the translation from the internal *TP53* promoter in intron 4 (affecting the production of Δ133p53 [34]) and the alternative splicing of intron 9 (affecting the production of p53β [34]).

In this context, tumours harbouring c.376-125T>C (rs9895829) or c.376-117G>A (rs35850753) variants in intron 4 showed increased Δ133p53 levels and, to a lesser extent, increased p53β levels (Figure 3H). This is in agreement with another study, which has demonstrated that rs9895829 alone or in combination with other *TP53* single nucleotide polymorphisms (SNPs) is associated with increased levels of Δ*133TP53* and *TP53β* (likely to be Δ*133TP53β*) and poor outcomes in glioblastoma and prostate cancer [50]. The authors proposed a possible explanation for the association between Δ133p53β generation and SNPs in *TP53,* where mutations in the internal promoter (intron 4) may affect the activity of the transcriptional machinery such as the elongation rate by polymerase II. This could affect cofactor binding such as transcriptional and splicing factors, and potentially facilitate the recruitment of splicing factors that favour retention of intron 9β [50]. Even though we have observed a significant association between variants in intron 4 and high levels of Δ133p53 when compared to the full-length protein, two tumours that harbour missense variants or a pathogenic variant (rs121912666) exhibited high TAp53 levels (Figure 3H). This suggests that a more comprehensive analysis of p53 status, instead of evaluations of specific SNPs, may provide more accurate conclusions regarding the relationship between *TP53* mutations and p53 isoform expression. Yet, the findings that p53β and Δ133p53 levels could be predicted by the presence of sequence variants are interesting given the mounting evidence associating Δ133p53β and cancer progression [33,44,47,80]. Hence, a predictive test could be developed by using these variants.

Similar to TAp53 in isolation, the effects of different p53 isoform composite patterns showed enrichment of *TP53* and *BRCA1* variants in high TAp53 and N-terminally truncated isoforms (either Δ133p53 or Δ160p53) and low p53β (Figure 4). In contrast, in tumours with high p53β, sequence variants in a breast cancer risk factor *CHEK2* were enriched [81]. Our recent studies have demonstrated the prognostic biomarker potential of the IHC p53 isoform profiles in breast cancer [64], more specifically of the KJC8 antibody (p53β) staining. Hence, the enrichment of genomic alterations in high p53β tumours supports the associations between p53β and worse cancer outcomes; however, given that the majority of samples presented a novel sequence variant in *CHEK2*, further functional assays are needed to assess the impact of this variant on protein functions. It should be noted that given that p53 isoform expression may be regulated post-transcriptionally and translationally, and that the RNA levels of these isoforms may not correlate with the protein product [64], a significant advantage of this work is that p53 isoform protein data was used instead of RNA levels. Nevertheless, the p53 isoform antibodies used in the IHC can detect multiple isoforms and detection can be affected by post-translational modifications.

Taken together, these results demonstrate that p53 levels predict breast cancer outcomes and diagnosis and are associated with *TP53* mutations. p53 isoform levels may be modulated by the presence of sequence variants in *TP53*, particularly Δ133p53 levels are associated with polymorphisms in intron 4 and alterations in genes of important signalling proteins such as *BRCA1*, *PALB2* and *CHEK2*. Nevertheless, the conclusions from this study must be tempered given that only a small number of samples in the cohort harboured common variants. Thus, confirmation of our findings in multiple and larger breast cancer cohorts is required.

## 4. Materials and Methods

### 4.1. Study Cohort

DNA extracted from 137 IDCs was provided by the Australian Breast Cancer Tissue Bank (Westmead, NSW, Australia). The cohort has previously been described [30,42,82]. This study was conducted in accordance with the Helsinki Declaration with ethical approval from the Hunter New England Human Research Ethics Committee (approval number: 09/05/20/5.02) and the University of Newcastle Health and Safety Committee (approval number: R7/2021). All patients agreed to the use of their clinical information and tissue for research.

### 4.2. Next-Generation Sequencing

DNA samples: DNA was checked for quality on an Agilent Tapestation with Agilent Genomic DNA Screen Tape and Reagents (Integrated Sciences, Chatswood, NSW, Australia) and all samples used in this analysis had a DNA integrity number of ≥7. DNA concentration was determined with the Qubit fluorometer using a dsDNA Broad Range Assay kit (ThermoFisher Scientific, Socresby, VIC, Australia) as per the manufacturer’s protocol. DNA samples were diluted to 12 ng/µL using Low TE buffer (Illumina, Northtech Singapore, Singapore) and stored at −80 °C until library preparation.

Library Preparation: NGS libraries were prepared from 60 ng genomic DNA (30 ng of input per pool, 2 pools) with an AmpliSeq PLUS kit (Illumina, Singapore), using an AmpliSeq Custom DNA Panel (Illumina, San Diego, CA, USA) for *TP53* and 19 gene-encoding p53 interactors (*HDM2, HDM4, PARP1, BARD1, ATR, RAD50, NBN, MYC, CDKN2A, H2AFX, ATM, CHEK1, BRCA2, RAD51, PALB2, BRIP1, PPM1D, BRCA1* and *CHEK2*; Appendix A; 93.39% coverage, number of amplicons = 747, amplicons < 275 bp) as per the manufacturer’s protocol (AmpliSeq for Illumina On-demand, Custom and Community Panels Reference Guide; Document # 1000000036408 v09). Libraries were checked for size on an Agilent Tapestation with Agilent D1000 Screen Tape and Reagents (Integrated Sciences) and concentration was determined with the Qubit fluorometer using dsDNA High Sensitivity Assay kit (ThermoFisher Scientific) as per the manufacturer’s protocol. Eight library samples were concentrated using a SpeedVac vacuum concentrator system (ThermoFisher Scientific) due to low library yields (<2 nM). Libraries were manually normalised to 2 nM, pooled, denatured and diluted to 1.9 pM.

Sequencing: pooled libraries were sequenced on an Illumina NextSeq 500 with a NextSeq 500/550 Mid Output Kit v.2.5 (300 cycles, paired end). The average sequencing depth was found to be 3293x and uniformity of coverage was >94%.

### 4.3. Bioinformatics

Alignment and variant-calling were performed using the DNA Amplicon App v2.1.1 (Illumina) using the manifest file IAA28008_182_manifest.txt (Appendix A), the *Homo sapiens* UCSC hg19 as the reference genome and Ensembl as the annotation source. The VCF files were further annotated using the vcf2maf utility [83] to produce mutation annotation format (MAF) files. The MAF files were then visualised using Maftools R-package [67]. Only variants with quality values of 100, in non-repeat regions (≤6 repeats), with frequency > 0.01 and low strand bias (<0.05), and coverage depth of ≥200 were considered for the analysis. The impact of sequence variants was assessed using Ensembl variant effect predictor (VEP) [84], PolyPhen [85] and SIFT [86].

Catalogue of Somatic Mutations in Cancer (COSMIC) [87], ClinVar [88], IARC TP53 [89] and VarSome [68] were used in combination with the annotated MAF to check the identified variants.

### 4.4. Immunohistochemistry of p53 Isoforms

To correlate *TP53* alterations with p53 and p53 isoforms’ protein expression, 108 immunohistochemically stained slides for p53 and p53 isoforms from our previous study (same cohort) were used [64]. Slides were scanned at 40× magnification using an Aperio AT2 scanner (Leica, Wetzlar, Germany), and analysed with HALO v3.3.2541 (Halo imaging analysis software, Indica Labs, Corrales, NM, USA) using the CytoNuclear v2.0.8 analysis mode.

### 4.5. Statistical Analysis

Mann-Whitney tests were performed for two comparisons such as the presence or absence of specific variants. Kruskal–Wallis test followed by Dunn’s multiple comparisons test was used to evaluate the relationship between p53 levels (weak, moderate, or strong H-scores) and tumour size, and to evaluate the relationship between specific variants and p53 isoform expression. The relationship between p53 levels and the triple-negative phenotype or the presence of high-impact variants was evaluated using Pearson’s chi-square test. The relationship between mutation and disease-free survival was performed using Kaplan-Meier analysis and comparison of survival curves was performed using the log-rank (Mantel–Cox) test. The enrichment of variants in specific genes in clinical features (hormone status) or in tumours expressing different levels of p53 isoforms (low versus high) was evaluated using Fisher’s exact tests. Statistical analyses were performed using Maftools R-package [67] and GraphPad Prism v. 9.0 (GraphPad Software, La Jolla, CA, USA). An adjusted *p*-value of <0.05 was considered statistically significant.

## Figures and Tables

**Figure 1 ijms-24-10078-f001:**
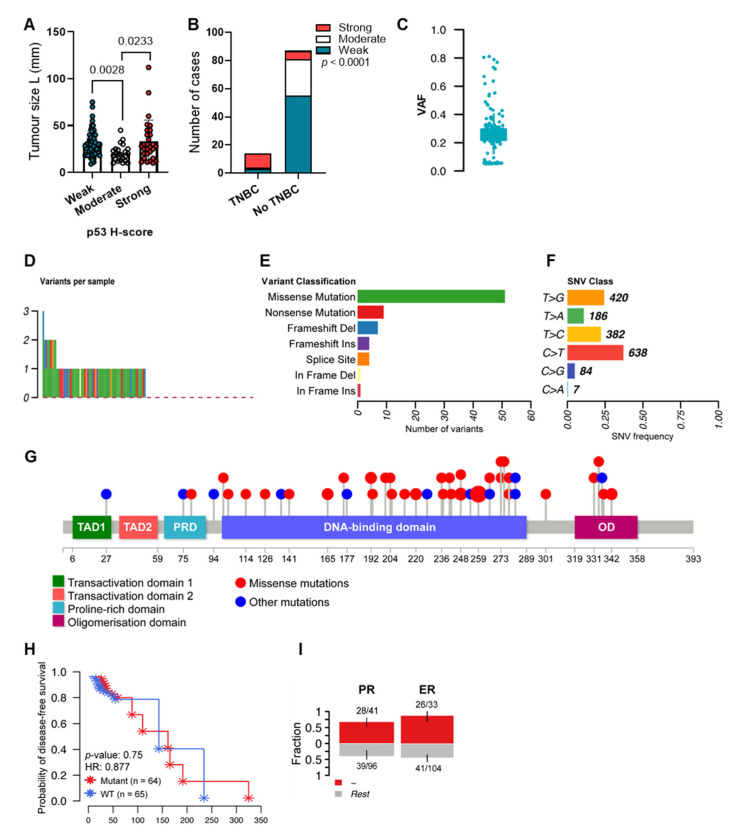
p53 levels and mutations are associated with clinicopathological outcomes in breast cancer. (**A**) Tumour size and (**B**) triple-negative phenotype of 108 IDCs from our previous study [64] stained for p53 (DO-1 antibody). H-scores were segregated into weak, moderate and high levels [65]. (**C**) VAF of *TP53* sequence variants detected in 137 IDCs. (**D**) Representation of the number of *TP53* variants per sample. Silent and c.559+2T>G variants were filtered out. Alterations were detected in 49% of samples. (**E**) Classification of *TP53* variants. (**F**) Frequency of different SNV classes. (**G**) Lollipop plot displaying mutation distribution and protein domains for *TP53* in IDCs. (**H**) Kaplan–Meier survival curve representing disease-free survival of cases distributed based on *TP53* mutation status into wild-type (WT) or mutated. HR: hazard ratio. (**I**) Bar plots displaying the association between *TP53* sequence variants and hormone status (negative or positive for ER or PR). Bars are annotated with the ratio of mutated samples to total samples. Error bars display 95% CI of binomial ratios. The *y*-axis denotes the fraction of samples harbouring *TP53* sequence variants. Kruskal–Wallis test followed by Dunn’s multiple comparisons test was used to determine the statistical significance of p53 levels and tumour size. The association between p53 levels and TNBC was evaluated using Pearson’s chi-square test. Log-rank (Mantel–Cox) test was used to determine the statistical significance of *TP53* mutation status and disease-free survival. The association between *TP53* status and hormone status was evaluated using Fisher’s exact test. Results were considered significant at *p* < 0.05.

**Figure 2 ijms-24-10078-f002:**
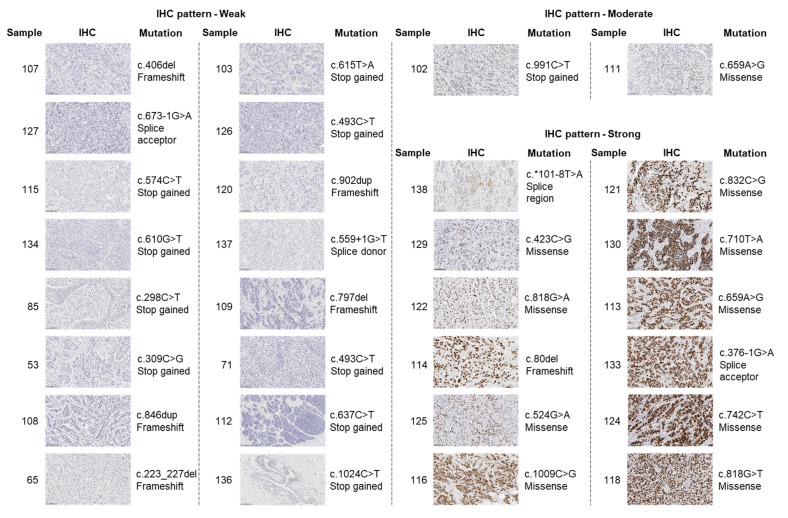
Mutations in *TP53* result in different levels of p53 in IDCs. Representative images of IDCs from our previous study [64] stained for p53 (DO-1 antibody). H-scores were segregated into weak, moderate and high levels [65]. Sample number is shown on the left and mutation details are shown on the right.

**Figure 3 ijms-24-10078-f003:**
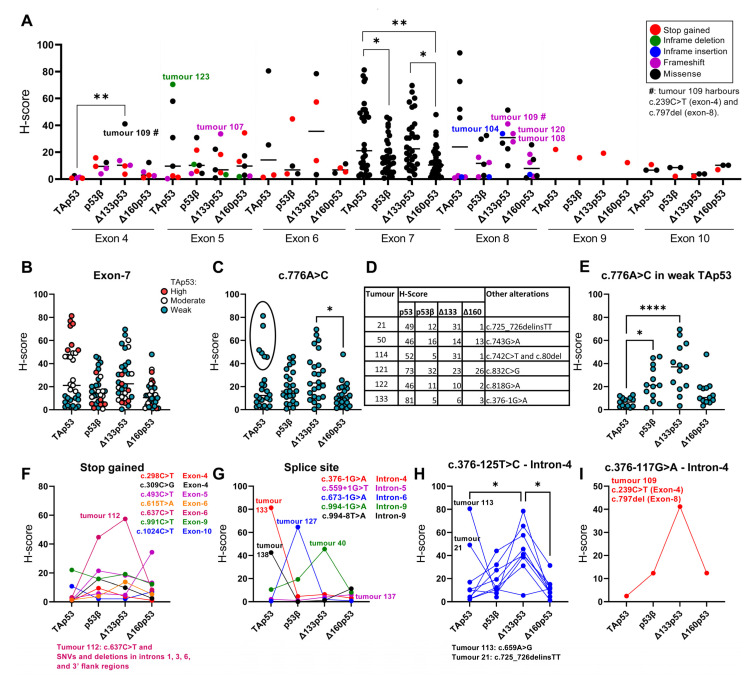
*TP53* mutation landscape and p53 isoform expression of IDCs. (**A**) Distribution of *TP53* mutations found in exons 4–10. The *y*-axis denotes the H-scores of p53 isoforms for mutated tumours. Distribution of p53 isoform levels in (**B**) samples with *TP53* missense mutations found in exon 7 (*n* = 31). TAp53 expression is coloured according to different levels. (**C**) Distribution of p53 isoform levels in samples harbouring a c.776A>C variant (*n* = 24). (**D**) Alterations detected in samples harbouring a c.776A>C variant and expressing high or moderate TAp53 levels (black circle in Figure 3C). (**E**) Distribution of p53 isoform levels in samples harbouring a c.776A>C variant and expressing weak TAp53 levels (*n* = 13). Distribution of p53 isoform levels in samples harbouring (**F**) stop-gained variants, (**G**) splice site variants, (**H**) c.376-125T>C variant, and (**I**) C.376-117G>A variant. Kruskal–Wallis test followed by Dunn’s multiple comparisons test was used to determine the statistical significance between the levels of distinct p53 isoforms. Results were considered significant at *p* < 0.05. * *p* < 0.05, ** *p* < 0.01, **** *p* < 0.0001. The notes next to the figures contain information on additional mutation sites of the respective tumours.

**Figure 4 ijms-24-10078-f004:**
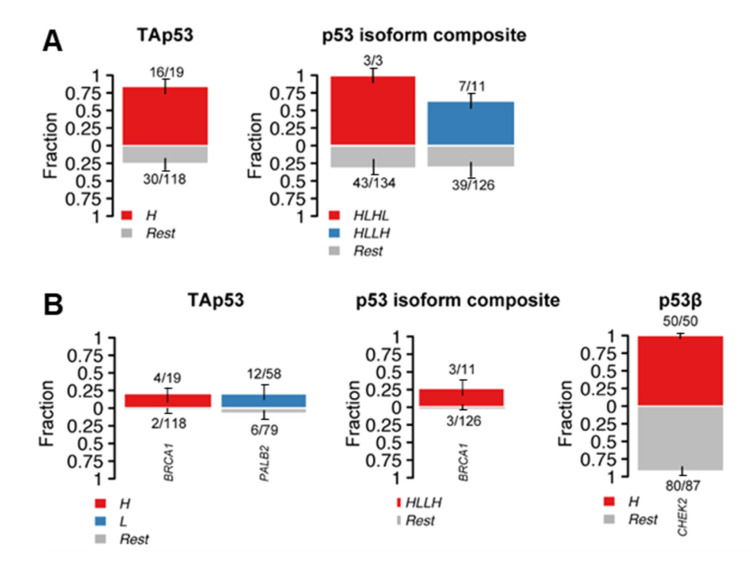
p53 and p53 isoform levels are associated with genetic alterations in breast cancers. (**A**) Bar plots displaying the association between *TP53* missense alterations and TAp53 levels (weak, moderate or high) or p53 isoform composite levels (low (L) or high (H)). (**B**) The association between alterations in genes of p53 interactors and TAp53 levels (weak, moderate or high) or p53 isoform composite levels (low (L) or high (H)), or p53β levels (low (L) or high (H)). Bars are annotated with the ratio of mutated samples to total samples. Error bars display 95% CI of binomial ratios. The *y*-axis denotes the fraction of samples harbouring missense sequence variants (**A**) or sequence variants in *BRCA1*, *PALB2* or *CHEK2* (**B**). Statistical analyses were performed using Fisher’s exact test. Results were considered significant at *p* < 0.05.

**Table 1 ijms-24-10078-t001:** High-impact *TP53* variants and expected outcomes.

Alteration			Tumours	IHC Pattern
Variant		Outcomes		
c.406del	deletion	-	107	Weak
c.673-1G>A	SNV	RNA splicing disruption and likely results in an absent or disrupted protein product.	127	
c.574C>T	SNV	Expected to result in loss of function by premature protein truncation or nonsense-mediated mRNA decay.	115 77 (IHC NA)	
c.610G>T	SNV	-	134	
c.298C>T	SNV	Predicted to cause nonsense-mediated decay and result in an absent protein.	85	
c.309C>G	SNV	Expected to result in loss of function by premature protein truncation or nonsense-mediated mRNA decay. This alteration is deficient at growth suppression.	53	
c.846dup	insertion	-	108	
c.223_227del	deletion	Novel variant.	65	
c.615T>A	SNV	Interpreted as a disease-causing mutation.	103	
c.493C>T		Expected to result in loss of function by premature protein truncation or nonsense-mediated mRNA decay.	126 71	
c.902dup	insertion	Expected to result in an absent or disrupted protein product.	120	
c.559+1G>T	splice donor	Expected to result in an absent or non-functional *TP53* protein.	137	
c.797del	deletion	-	109	
c.637C>T	SNV	Predicted to cause a truncated or absent *TP53* protein due to nonsense mediated decay.	112	
c.1024C>T	SNV	Predicted to cause a truncation of the encoded protein or absence of the protein due to nonsense mediated decay.	136 131 (IHC NA)	
c.991C>T	SNV	Prediction tools suggest that this variant may not impact RNA splicing.	102	Moderate
c.659A>G	SNV	Computational prediction suggests that this variant may have deleterious impact on protein structure and function.	111	
c.*101-8T>A	SNV	-	138	Strong
c.423C>G	SNV	Reported to have loss of transactivation capacity in yeast-based assays and is predicted to affect several p53 isoforms.	129	
c.818G>A	SNV	Computational prediction suggests that this variant may have deleterious impact on protein structure and function.	122	
c.80del	deletion	Expected to result in an absent or disrupted protein product.	114	
c.524G>A	SNV	A well-characterised mutation “hotspot” in the functionally critical DNA-binding domain.	125	
c.1009C>G	SNV	-	116	
c.832C>G	SNV	Reported to have loss of transactivation capacity in functional studies in yeast and human cell lines.	121	
c.710T>A	SNV	-	130	
c.659A>G	SNV	Computational prediction suggests that this variant may have deleterious impact on protein structure and function.	113	
c.376-1G>A	SNV	Expected to disrupt RNA splicing and lead to a loss of protein function.	133	
c.742C>T	SNV	Computational prediction suggests that this variant may have deleterious impact on protein structure and function.	124	
c.818G>T	SNV	A well-characterised mutation “hotspot” located within the functionally critical DNA-binding domain.	118	
c.1007del	deletion	-	91	NA
c.282del	deletion	-	135	
c.341dup	insertion	-	18	
c.602T>A	SNV	-	105	
c.681dup	insertion	-	128	
c.760_761del	deletion	-	106	
c.994-1G>A	SNV	This variant induces altered splicing and may result in an absent or disrupted protein product.	40	

SNV: single nucleotide variant; NA: not available.

**Table 2 ijms-24-10078-t002:** *BRCA1*, *CHEK2,* and *PALB2* sequence variants detected in 137 IDCs. Silent sequence variants were filtered out using MAF tools [67].

Gene	Variant	dbSNP	Impact	N *	p53 Isoform Expression **
** *BRCA1* **	p.Ile562Thr	rs1555591375	Moderate	1	High TAp53 and Δ160p53
	p.Ser768Arg	novel	Moderate	1	
	p.Ala194Val	novel	Moderate	1	
	p.Asp295His	novel	Moderate	1	
	p.Glu881Ter	rs397508988	High	1	NA
	p.Glu1570IlefsTer2	novel	High	1	
	p.Asp219His	rs273902779	Moderate	1	
** *CHEK2* **	p.Thr421Pro	novel	Moderate	130	NA (*n* = 35) High p53β (*n* = 50) Low p53β (*n* = 45)
	p.Val146Phe	rs907433465	Moderate	1	High p53β
	p.Val25Ile	rs142243299	Moderate	1	NA
** *PALB2* **	p.Tyr551Asn	novel	Moderate	13	NA (*n* = 2) High TAp53 (*n* = 3) Low TAp53 (*n* = 5)
	p.Thr721Ile	novel	Moderate	1	High TAp53
	p.Thr193AsnfsTer2	rs875989790	High	1	
	p.Ser779Ter	rs764509489	High	1	Low TAp53
	p.Leu100Phe	rs61756147	Moderate	1	
	p.Leu1143ThrfsTer14	rs587776425	High	1	NA
	p.Glu384Ter	-	High	1	

* Number of specimens harbouring the variants. ** p53 isoform expression from the enrichment analysis (Figure 4). NA: not available.

## Data Availability

Sequencing data have been deposited in a publicly available dataset (Sequence Read Archive): BioProject accession number: PRJNA971728. Other data is available from the corresponding author upon reasonable request.

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
