# Peer review of "p53 Dysregulation in Breast Cancer: Insights on Mutations in the TP53 Network and p53 Isoform Expression"

_ijms, 2023, doi:10.3390/ijms241210078_

Round 1
Reviewer 1 Report
The analytic results presented in the manuscript suggest the association between TP53 variants and the expression of P53 isoforms in the breast cancer samples. Although bench work analyses, such as sited-directed mutations, are needed to test the suggested association, the current work contributes to the understanding of molecular mechanisms of TP53 variant in cancer progression.
Author Response
We would like to thank the reviewer for their positive feedback.
Reviewer 2 Report
This paper was tried to assay the NGS data with p53 mutations (missense, nonsense, deletion….) to compare the IHC of p53 isoforms expression. All results are complex and need to organize and clarify to make the results much clear.
1. Previous report showed that hot spot mutants of p53 can find the enhancement of shorter p53 isoforms [EMBO Reports (2016)17:1542-1551]. But this study showed that most of the missense mutations cannot find the p53 isoforms with changes. Please try to pickup the top 3 or Top 5 of p53 hot spot mutants of p53 in breast cancer (the candidates can be found on COSMIC database), and assay the p53 isoforms expression pattern of your clinical samples.
2. Since most of the sample with high p53 isoforms expression is due to contain more than one site of mutation, but these key information cannot be found on the spot(s) in each figure and just mentioned in the main article. It should be needed to label as the note on the side of spot or added a table following the figure.
Round 2
Reviewer 2 Report
The revision of the article is now much clear. But I still have one question about Figure 4B. Why almost all the samples with CHEK2 variants ? I picked up form COSMIC database showed that CHEK2 with no so much high mutation rate (only 80 mutants /3297 total in breast cancers) vs (1325/16185 in all type of cancer). It should be carefully addressed the sequencing again. If all the Fig 4B each subgroup all are less than 12 case sequencing variants (at least BRCA1 and PALB2), please give us the all detail sequencing alterations (maybe a table) in all the sample mentioned in Fig 4B.
Round 3
Reviewer 2 Report
The new provided information about BRAC1, CHEK2 and PALB2 variants is useful, and the revision is now suitable for publication.